# Pathophysiological profile of non-ventilated lung injury in healthy female pigs undergoing mechanical ventilation
Elena Spinelli[1], Anna Damia[2], Francesco Damarco[3], Beatrice Gregori[2], Federica Occhipinti[2], Zara Busani[2], Marco Leali[2], Michele Battistin [4], Caterina Lonati [4], Zhanqi Zhao[5], Alessandra Maria Storaci[2,6], Gianluca Lopez[6], Valentina Vaira[2,6], Stefano Ferrero[6,7], Lorenzo Rosso [2,3], Stefano Gatti [4] & Tommaso Mauri [1,2] ✉

## Abstract

**Background** Lung regions excluded from mechanical insufflation are traditionally assumed to be spared from ventilation-associated lung injury. However, preliminary data showed activation of potential mechanisms of injury within these non-ventilated regions (e.g., hypoperfusion, inflammation).

**Methods** In the present study, we hypothesized that non-ventilated lung injury (NVLI) may develop within 24 h of unilateral mechanical ventilation in previously healthy pigs, and we performed extended pathophysiological measures to profile NVLI. We included two experimental groups undergoing exclusion of the left lung from the ventilation with two different tidal volumes (15 vs 7.5 ml/kg) and a control group on bilateral ventilation. Pathophysiological alteration including lung collapse, changes in lung perfusion, lung stress and inflammation were measured. Lung injury was quantified by histological score.

**Results** Histological injury score of the non-ventilated lung is significantly higher than normally expanded lung from control animals. The histological score showed lower intermediate values (but still higher than controls) when the tidal volume distending the ventilated lung was reduced by 50%. Main pathophysiological alterations associated with NVLI were: extensive lung collapse; very low pulmonary perfusion; high inspiratory airways pressure; and higher concentrations of acute-phase inflammatory cytokines IL-6, IL-1β and TNF-α and of Angiopoietin-2 (a marker of endothelial activation) in the broncho-alveolar lavage. Only the last two alterations were mitigated by reducing tidal volume, potentially explaining partial protection.

**Conclusions** Non-ventilated lung injury develops within 24 h of controlled mechanical ventilation due to multiple pathophysiological alterations, which are only partially prevented by low tidal volume.

## Plain language summary

Respiratory failure that occurs in cases of atelectasis, pneumonia and acute hypoxemic respiratory failure a machine called a mechanical ventilator is used to move air in and out of the patient's lungs. We know that the use of a mechanical ventilator can induce lung injury, but complete exclusion from ventilation might not be safe. Using pig lungs to mimic the patient's lungs, we evaluated the use of a ventilator against non-use. We find that the lungs sustained injury regardless of ventilator use. The non-ventilated lung injury consisted of collapse (lack of expansion), low amount of blood flow, high ventilation pressure and inflammatory response. Physicians should be aware that also the regions of the lung not receiving ventilation are at risk of injury.

Loss of airway patency (airway closure) excludes part of the lungs from tidal ventilation, inducing alveolar collapse. Presence of non-ventilated atelectatic lung regions is quite common in patients with acute respiratory failure, as they may derive from multiple causes such as secretions obstructing the bronchial tree, negative transpulmonary pressure and/or superimposed lung edema compressing the airways.

Presence of non-ventilated lung regions decreases the end-expiratory aeration and increases the regional tidal volume in ventilated areas, increasing regional stress and strain and reducing compliance[1]. Hypoxic pulmonary vasoconstriction develops in the non-ventilated areas to divert the blood flow to the ventilated regions[2], which become at increased risk of over-perfusion edema[3]. Thus, in the presence of non-ventilated lung

---

regions, the residual ventilated areas are at higher risk of developing ventilator-induced lung injury (VILI)[1].

Given the lack of stretch induced by high airway pressure and tidal ventilation, the non-ventilated lung has traditionally been considered self-protected from injury to the point that some authors recommend protecting the lung by keeping it "closed"[4]. Pilot data, instead, suggest that presence of non-ventilated lung areas could trigger local detrimental processes[5]: for example, prolonged loss of aeration progressively activates inflammatory gene expression[6]; short-term one-lung ventilation increases markers of ischemic injury to the non-ventilated side[5,7,8]; excessive stress to the residual ventilated areas could trigger diffuse lung inflammation[9,10].

In the present study, we study healthy pigs with one lung intentionally excluded from mechanical ventilation for 24 h, while the other undergoes different ventilation strategies. The non-ventilated lung develops injury (Non-Ventilated Lung Injury - NVLI) associated to hypoperfusion, collapse, high inspiratory pressure, and inflammation; decreased tidal volume with lower inspiratory pressure attenuates some of these mechanisms, resulting in incomplete protection from NVLI.

## Methods

### Ethics
The study was approved by the Italian Ministry of Health, Rome, Italy (Auth. No. 246/2022-PR, Protocol No. 568EB.34 (ex 32)) and conducted according to the European Directive 2010/63/EU on the protection of animals used for scientific studies and the Italian decree 26/2014. Approval by the Institutional Committee for Animal Care was obtained before starting the experiments.

### Animal preparation
Twenty-seven healthy female pigs (Sus scrofa, 35 ± 5 Kg) were anesthetized, tracheostomized, and instrumented with invasive hemodynamic monitoring. Electrical Impedance Tomography (EIT) belt was positioned at mid chest position and connected to its monitor.

### Study protocol
After preparation and check for healthy baseline condition, supine animals were randomly allocated to one of the following study groups:

- NVLI-15 group ($n = 10$), receiving mechanical ventilation with exclusion of the left lung by dual-lumen tube. Mechanical ventilation settings were: volume-controlled mode, $V_T$ 15 mL.kg$^{-1}$, RR 15 bpm, I:E 1:2, PEEP 1 cmH$_2$O (i.e., the lowest to guarantee optimal pneumatic performance by the ventilator), FiO$_2$ 0.5
- NVLI-7.5 group with lower $V_T$ and permissive hypercapnia ($n = 11$), receiving mechanical ventilation with exclusion of the left lung by dual-lumen tube. Same ventilation settings and FiO$_2$ as for NVLI-15 group, apart from $V_T$ 7.5 mL.kg$^{-1}$
  Tidal volumes of 15 and 7.5 mL.kg$^{-1}$ were selected to target moderate and minimal lung injury to the ventilated lung, respectively, based on previous publications[11,12]
- MV-Control group with bilateral mechanical ventilation ($n = 6$) by single-lumen tube and the same settings and FiO$_2$ as for NVLI-15 group

Fluids were administered by prespecified protocol, targeting stable arterial blood pressure and neutral to slightly negative fluid balance.

Additional description of the study protocol can be found in Supplementary Methods.

### Study measures
Physiological data (see below) were collected after 2, 6, 12, 18 and 24 h (T2, T6, T12, T18, T24) from start of ventilation.

At T24, after collection of the physiological measures, we performed right and left selective broncho-alveolar lavages (BAL) with 30 ml 0.9% saline solution per side. Then, animals were euthanized, and 3 tissue samples from each lung (basal, middle, apex) were collected.

### Development of NVLI
Blinded expert pathologist assessed the severity of injury for each lung by validated composite histological score, with 10 sub-items ranging from 0–3 (total score range: 0–30), as previously described[13]. Values from the 3 samples from each side were averaged to obtain the representative left and right lung scores. Wet-to-dry lung weight ratios were measured for each side, too.

### Pathophysiological differences between study groups
Physiological data collected at each time point were compared between groups to highlight potential mechanisms associated with NVLI. We collected: respiratory mechanics including transpulmonary pressure by inspiratory and expiratory occlusions; arterial and mixed venous blood gas analyses; systemic and pulmonary hemodynamics; cumulative fluid balance; and EIT perfusion data. EIT perfusion maps were derived from offline analysis of the time-impedance curve obtained by first pass of a 10 ml-bolus of 5% saline solution injected during an end-inspiratory occlusion, as previously described[14,15]. Perfusion of the non-ventilated left lung was quantified as the fraction of blood flow perfusing the non-ventilated pixels in the left hemithorax.

### Inflammation
At T24, BAL fluids recovered from each lung were centrifugated, the supernatants promptly stored at −80 °C and, later, separately assayed by ELISA for concentrations of acute inflammatory mediators (IL-6, IL-1β and TNFα)[16] and of markers of inflammatory activation of the alveolar epithelium vs. endothelium (i.e., Surfactant Protein D (SP-D) and soluble Receptor for Advanced Glycation End-products (sRAGE) vs. Angiopoietin 2 (Ang2))[17–19].

### Statistics and reproducibility
Data are shown as mean ± standard deviation or median [interquartile range], as appropriate. Data between groups were compared using one-way ANOVA or Kruskal-Wallis test, followed by Holm-Sidak or Dunn's test for multiple comparisons. Longitudinal data (physiological and EIT variables along the study timepoints) were analyzed using repeated measures two-way ANOVA, with timepoints and study groups as covariates. $P$-value < 0.05 indicated statistical significance. Analyses were performed using GraphPad Prism 9 (GraphPad Software, San Diego, CA).

### Sample size
Difference in the histological score of the left lung between the NVLI-15 vs. MV-Control group was the primary endpoint of the study. Sample size was comparable to previous animal studies on similar topic[11,14]. We also performed a power analysis by hypothesizing, based on previous studies[14,15], values of the histological scores of 10.5 ± 3.0 vs. 5.0 ± 3.0: to obtain power of 0.8 with alpha 0.05. The minimum sample size resulted in 5 animals per group.

### Reporting summary
Further information on research design is available in the Nature Portfolio Reporting Summary linked to this article.

## Results

### NVLI develops within 24 h of mechanical ventilation
The histological injury score of the left lung was higher in the NVLI-15 group, compared to MV-Control ($p < 0.001$). Histological score of the NVLI-7.5 was high-intermediate: lower than NVLI-15, but higher than MV-Control groups (Fig. 1a). The following histological sub-scores differed between groups, with highest values in the NVLI-15 group: alveolar neutrophils infiltration, alveolar macrophage proliferation, interstitial congestion, alveolar hemorrhage, interstitial lymphocytes proliferation (Fig. 1c).

Animals in the NVLI-15 group also presented higher values of right lung histological injury score compared to NVLI-7.5 and MV-Control groups, which were similarly low ($p < 0.01$) (Fig. 1b).

The histological score of the left lung was higher than the right one for both NVLI groups ($p < 0.001$), suggesting that the severity of NVLI could be higher than the associated VILI to ventilated regions.

Figure 2 shows representative histological samples from right and left lung for the 3 study groups: left lungs of the NVLI groups were characterized by extensive alveolar collapse.

Left lung wet to dry ratios were low and didn't differ between study groups (Supplementary Fig. 1), probably due to left side hypoperfusion in NVILI groups (see below), limiting the amount of fluids that could exudate.

## Pathophysiological variables associated with the development of NVLI

Longitudinal analysis of EIT data showed hypoperfusion of the left lung (down to around 20% of total pulmonary perfusion, corresponding to roughly 0.5 l/min) in both NVLI groups, in comparison to MV-Control (Fig. 3a and Supplementary Fig. 2). Regional ventilation and perfusion distribution measured by EIT in representative animals from each study group are shown in Fig. 4.

Different tidal volumes between NVILI groups affected static and dynamic inspiratory pressure applied to the ventilated right lung: driving transpulmonary and plateau pressure, indeed, were higher in the NVLI-15 group as compared to the NVLI-7.5 and MV-Control groups (Fig. 3b–c).

As expected, arterial $CO_2$ levels were higher and pH lower in the NVLI-7.5 group, leading to permissive hypercapnia with mild respiratory acidosis for the first 12–18 h (Fig. 3d).

Fluid balance was slightly negative and lower in the NVLI-7.5 group, as compared to similar neutral values in the other 2 groups (Fig. 5a), but this likely didn't affect study primary endpoint, as lung edema wasn't a main feature of left lung injury (see above). Pulmonary vascular resistance showed higher values in all NVLI groups, increasing mean pulmonary artery pressure (PAP) in comparison to MV-Control (Fig. 5b–c). However, given the inhomogeneous distribution of pulmonary perfusion, the injurious effects of high PAP should have affected mainly the right lung, again having minimal influence on the study primary endpoint.

Longitudinal values along all study timepoints of the remaining physiological variables are reported in Supplementary Data 1.

## NVLI is associated to acute inflammatory activation, modulated by tidal volume

Levels of all 3 acute phase inflammatory cytokines (i.e., IL-6, TNF-α and IL-1β) in BAL collected from the left lungs were higher in the NVLI-15 vs. NVLI-7.5 and MV-Control groups (Fig. 6a–c). The same was true for levels in BAL recovered from the right lungs (Fig. 7a–c).

To discriminate contribution by endothelial activation as opposed to epithelial dysfunction, BAL concentrations of Ang2, SP-D and sRAGE were measured (Fig. 6c–e). Similarly to acute phase cytokines, levels of the endothelial inflammatory marker Ang2 were higher in the left lung of the NVLI-15 group, as compared to the NVLI-7.5 and MV-Control groups, while epithelial markers didn't differ between groups (Fig. 6c–e). The same results were confirmed for right lungs BAL (Fig. 7c–e).

## Discussion

The main findings of this experimental study can be summarized as follows: non-ventilated lung regions develop injury (NVILI) within 24 h of

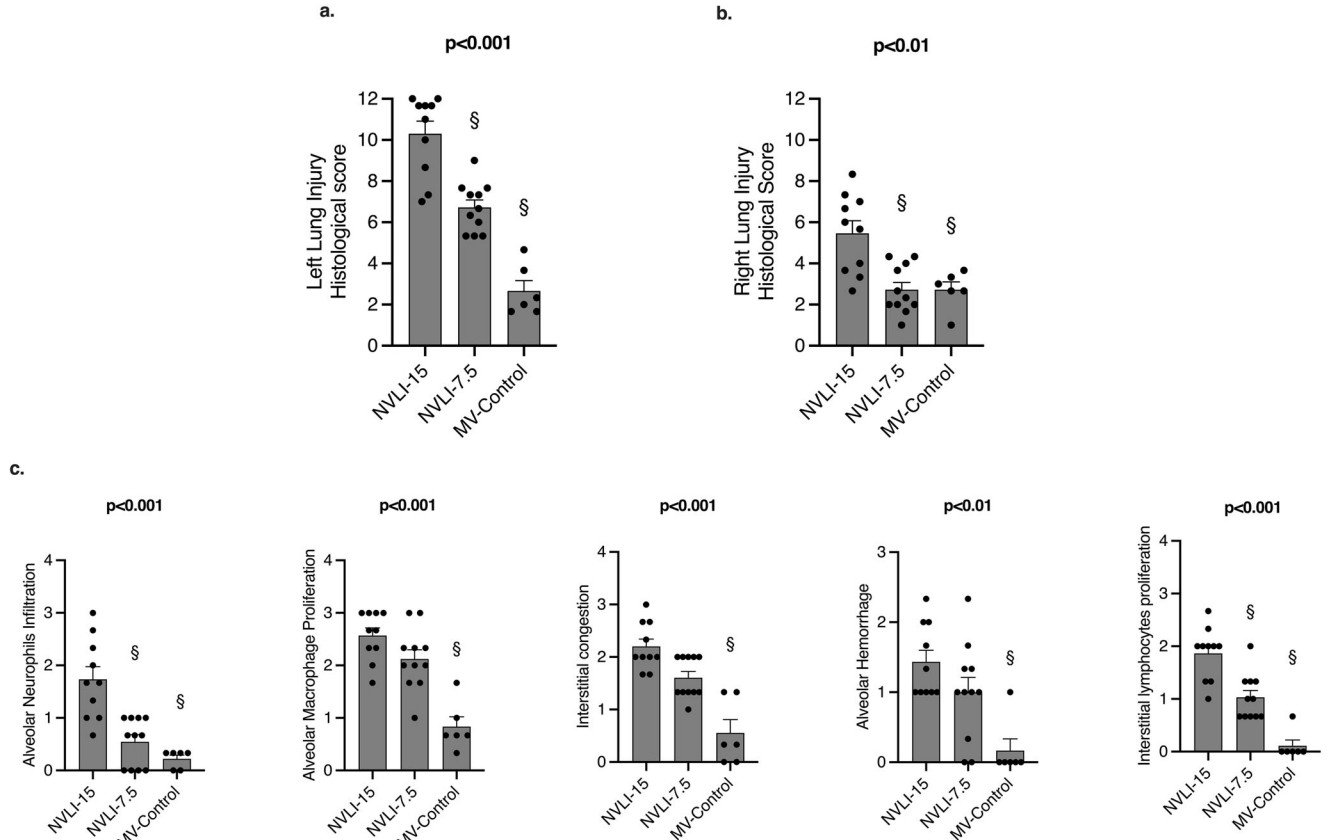

**Fig. 1 | Lung injury histological scores.** Left lung histological score showed significantly higher values in NVLI-15 group in comparison to NVLI-7.5, which, however, were higher than MV-Control group (**a**). Right lung histological score showed significantly higher values in the NVLI-15 group in comparison to NVLI-7.5 and MV-Control groups (**b**). Left lung histological sub-scores including alveolar neutrophils infiltration, alveolar macrophage proliferation, interstitial congestion, alveolar hemorrhage, interstitial lymphocytes proliferation showed higher values in the NVLI-15 group (**c**). Data are expressed as scatter plot with bars and error bars (mean ± SEM). Comparisons were performed by Kruskal-Wallis test for non-normally distributed values (*P*-value reported in the graph) followed by Dunn's multiple comparisons test ($^\S p < 0.05$ vs. NVLI-15 group). Sample size: NVLI-15 group $n = 10$; NVLI-7.5 $n = 11$; MV-Control $n = 6$.

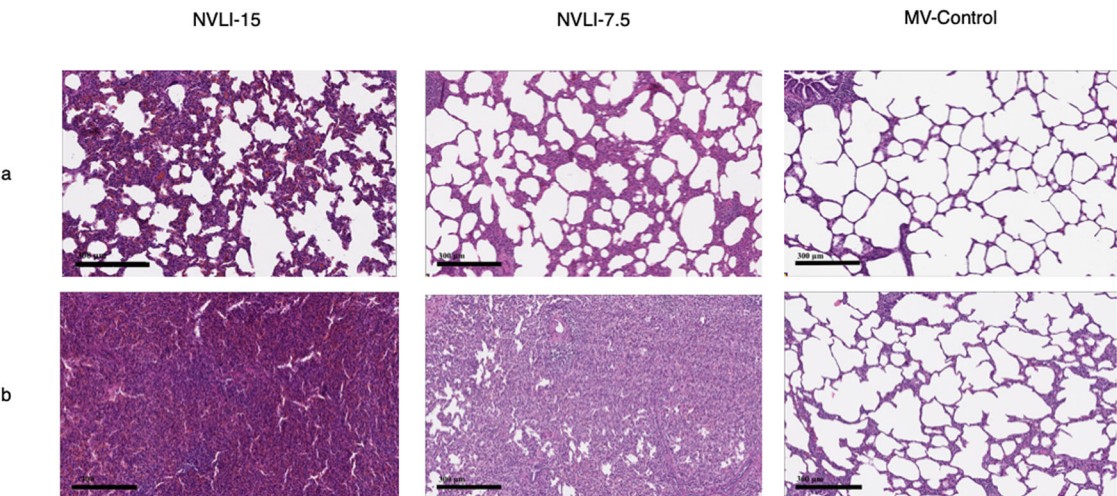

**Fig. 2 | Microscopic appearance of the lungs at histological analysis.** Representative microphotographs of the right (**a**) and left (**b**) lungs from the NVLI-15 group, NVLI-7.5 group and MV-Control group. Note the collapse characterizing both NVLI groups. Hematoxylin-Eosin stain was used and the scale bar of 300 μm applies to all the images.

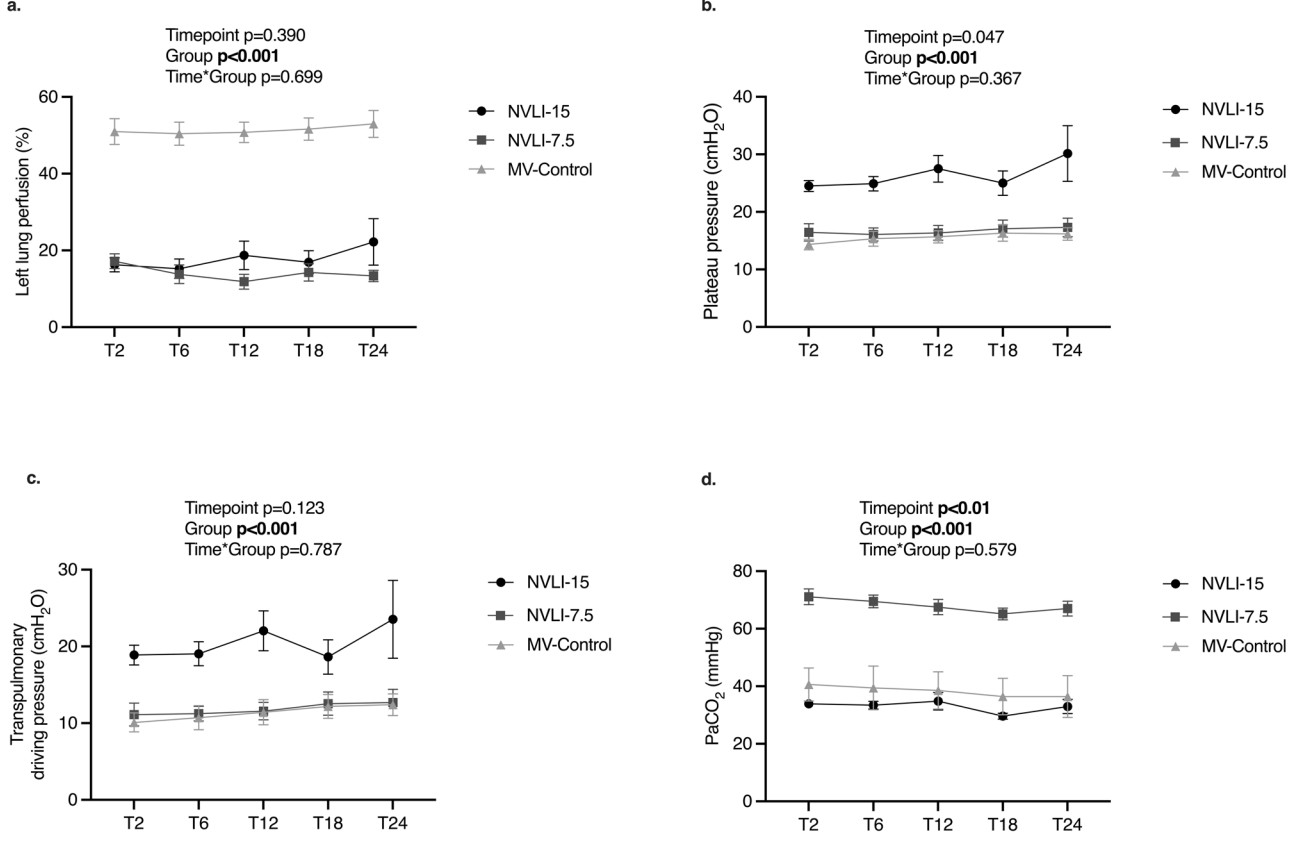

**Fig. 3 | Fraction of pulmonary perfusion reaching the left lung, static and dynamic lung stress and arterial partial pressure of $CO_2$ along all study timepoints.** EIT-measured fraction of pulmonary perfusion reaching the left lung (**a**) was minimal in all NVLI groups at all timepoints and homogenous around 50% in the MV-Control group. Stress imposed onto the right lung was higher in the NVLI-15 group as compared to the other two study groups: plateau pressure was considered as a measure of static stress (**b**) and transpulmonary driving pressure as a measure of dynamic stress (**c**). Arterial $CO_2$ levels, as expected, were higher in the NVLI-7.5 group compared to all other study groups (**d**). Data in **a**–**d** are expressed as mean ± SEM. Comparisons are obtained by mixed-effect model for repeated measurements. *P*-values are reported in the graph. Sample size: NVLI-15 group $n = 10$; NVLI-7.5 $n = 11$; MV-Control $n = 6$.

mechanical ventilation; the severity of NVILI at 24 h is reduced by decreased tidal volume and inspiratory pressure; mechanisms associated with NVILI could include regional hypoperfusion and collapse, higher lung stress to the ventilated regions, and diffuse inflammatory activation with contribution by the alveolar endothelium. Among these, decreased tidal volume could not prevent hypoperfusion and collapse affecting the non-ventilated regions.

Our experimental model aimed to mimic, in a more schematic and easier to dissect fashion, the coexistence of non-ventilated (i.e., excluded

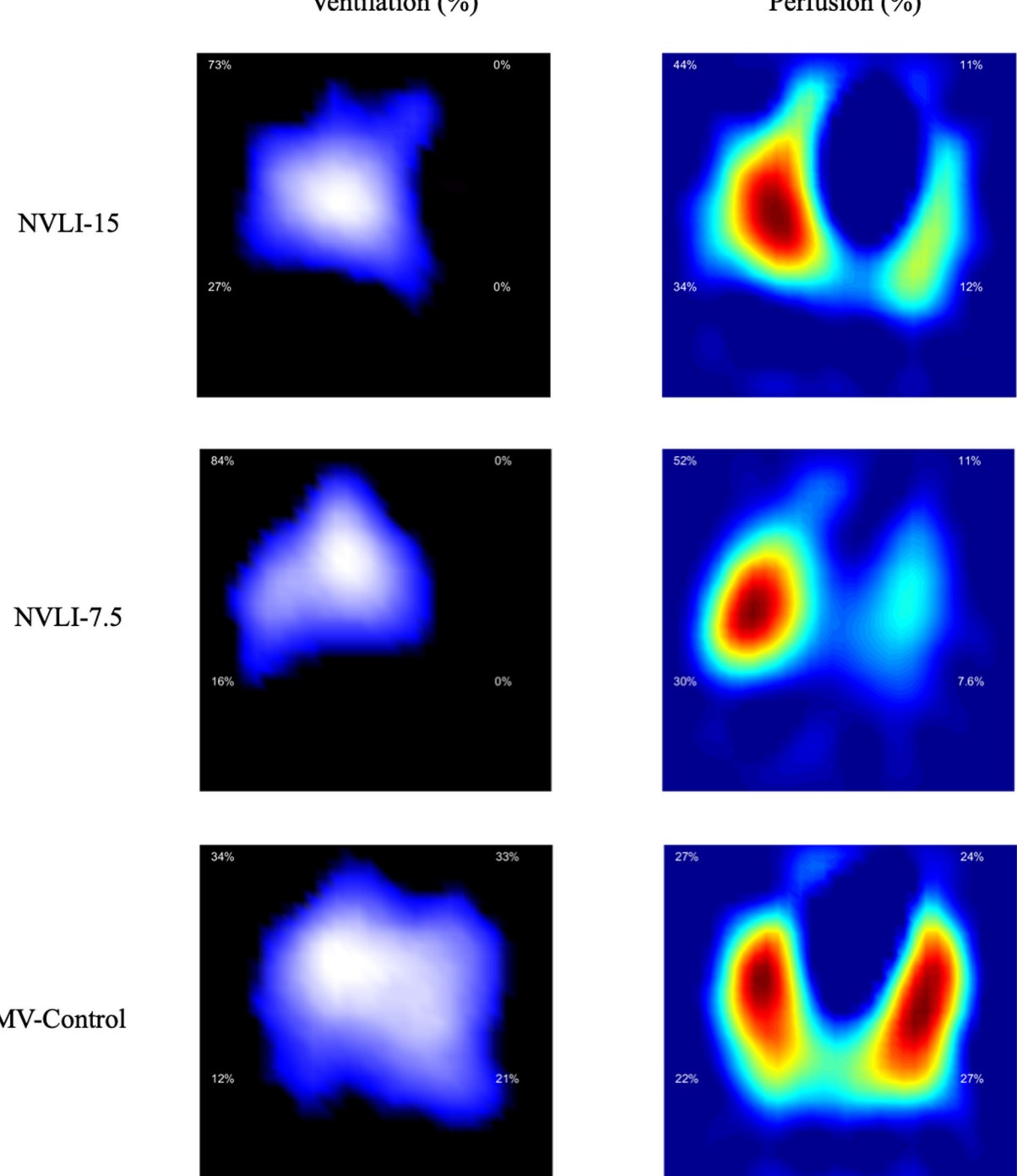

**Fig. 4 | Distribution of ventilation and perfusion by EIT.** EIT-based images show normal homogenous bilateral distribution of ventilation and perfusion in the MV-Control group vs. unilateral ventilation and low comparable values of left lung perfusion in the NVLI-15 and NVLI-7.5 groups. Data are expressed as % of lung ventilation and perfusion reaching each lung region. The right lung is showed on the left, while the left lung is showed on the right. Sample size: NVLI-15 group $n = 10$; NVLI-7.5 $n = 11$; MV-Control $n = 6$.

and progressively de-aerated, compressed) lung regions with normally- and over-ventilated and -perfused ones, as occurs in several acute pulmonary conditions, including, for example, atelectasis, pneumonia and acute hypoxemic respiratory failure.

It is recognized that presence of non-ventilated regions promotes VILI[20], because they induce overdistension to the residual ventilated lung and they increase shear stress. We confirmed that higher tidal volume reaching the ventilated lung induces higher static and dynamic stress, ultimately leading to more severe lung injury. As exclusion from ventilation is a dynamic condition which, at early stage, may not correspond to lung collapse, it could be relevant for prevention of VILI to recognize patients with larger non-ventilated lung regions by dedicated dynamic monitoring (e.g., EIT)[21,22].

On the other hand, whether exclusion from ventilation *per se* exerts an injurious local effect is a matter of debate. Non-ventilated regions have traditionally been thought to be spared from injury[4,23], as they are not exposed to dynamic stretch and atelectrauma[24,25]. Indeed, limiting ventilation (and increasing dynamic collapse) could decrease injury and

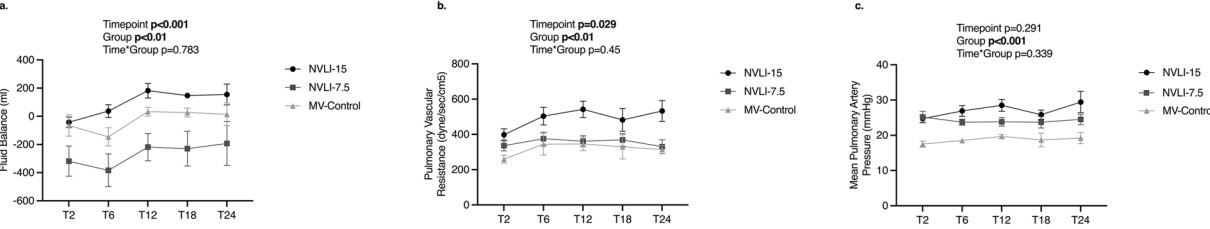

**Fig. 5 | Fluid balance, pulmonary vascular resistance and mean pulmonary artery pressure along all study timepoints.** Fluid balance (panel **a**) was slightly negative and lower in the NVLI-7.5 group, as compared to similar neutral values in the other 3 groups. Pulmonary vascular resistance (PVR) (**b**) showed higher values in all NVLI groups, increasing mean pulmonary artery pressure (mPAP) (**c**) in comparison to MV-Control group. Data in (**a–c**) are expressed as mean ± SEM. Comparisons are obtained by mixed-effect model for repeated measurements (results from the Tukey's multiple comparison test can be found in Supplementary Data 2 online). P-values are reported in the graph. Sample size: NVLI-15 group $n = 10$; NVLI-7.5 $n = 11$; MV-Control $n = 6$.

inflammatory spread as compared to high-stretch ventilation[24]. However, non-ventilated regions are potentially exposed to the detrimental effects of regional hypoperfusion and alveolar collapse. Decreased alveolar oxygen (due to lack of ventilation) combined with reduction of oxygen delivery (due to hypoxic pulmonary vasoconstriction) could promote inflammation in non-ventilated regions. Experimental studies in rats showed that alveolar hypoxia induces an inflammatory response with recruitment of macrophages in the lung[26]. In addition, hypoxia-induced inflammation with accumulation of neutrophils has been described in non-ventilated atelectatic lungs[27]. This is in line with our result showing increased neutrophils infiltration in the left lung of NVLI-15 group. Alveolar collapse, due to lack of tidal ventilation with rapid aeration loss, has also been associated with increased inflammation, even when hypoperfusion is prevented[28]. Therefore, previous studies and our results indicate that exclusion from ventilation induces multiple local alterations which affect the regulation of the inflammatory response and the function of the alveolar-capillary barrier[6], with potentiation of the neutrophil response[27,29]. In our experiment, 24 h might have been enough to further show progression from activation of inflammation to tissue injury.

The alveolar endothelium is a key player in the activation of lung inflammation, especially when mechanisms of injury originate from the vascular side[23]. In this context, Ang2 has been described as a biomarker with autocrine functions, activated by endothelial stress and inflammation and released by the alveolar endothelium[30]. Ang2 further amplifies the regional inflammatory activation, inducing angiogenesis, alveolar-epithelial barrier dysfunction and worsening lung injury[31]. In the present study, we detected higher levels of Ang2 in both lungs of NVILI-15 group, thus suggesting diffuse inflammatory activation of the endothelium during NVLI. Decreased levels of Ang2 associated with lower tidal volumes could indicate excessive production triggered by higher lung stress or lower degree of NVLI due to lower duration of collapse (see below). Markers of epithelial dysfunction, instead, were not increased, thus underlying potential differences between NVLI and classical VILI.

In the present study, the severity of local injury to the non-ventilated regions was partially reduced by reduced tidal volume reaching the ventilated lung. Lowering tidal volume decreased inspiratory stress to the ventilated regions and inflammation in both lungs, without affecting left lung hypoperfusion and collapse. The association of a larger number of detrimental mechanisms could explain the highest degree of injury in the animals of NVLI-15 group (4 mechanisms of injury) compared to intermediate values for the NVLI-7.5 (2 mechanisms) and no injury in the MV-Control (no mechanisms). Marked increase in the alveolar levels of the inflammatory cytokines IL-6, IL-1β and TNF-α and of Ang2 were evident in both ventilated and non-ventilated lungs of the NVLI-15 group. This might suggest that cross-talk from the stressed ventilated lung could have exacerbated NVLI-associated inflammation, or that larger tidal volume might have accelerated compression and timing to complete collapse of the left lung, enhancing local inflammation which later spread to the ventilated lung. Indeed, IL-6 and TNF-α have previously been identified as biomarkers of VILI in the broncho-alveolar of ventilated lungs after single side

ventilation[32] and, in the NVLI-7.5 group, levels of alveolar cytokines were dampened in both lungs. While it is recognized that mediators released by the overdistended lung could hit distant organs[33], spreading of inflammation between different regions of the lung (or between the two lungs) is physiologically plausible, but difficult to demonstrate. We could hypothesize that lower inspiratory stress dampened severity of NVLI by preventing a second inflammatory "hit" to the non-ventilated lung. Similarly, systemic inflammation induced by lipopolysaccharide exacerbates injury in the atelectatic lung undergoing mechanical ventilation[28]. Alternatively, lower tidal volume might have slowed time to complete collapse and the exposure to its detrimental consequences. Finally, permissive hypercapnia in the NVLI-7.5 group could have, at least theoretically, contributed to the partial protection from NVLI[34], possibly also explaining the lower levels of inflammatory cytokines in the NVLI-7.5 as compared to MV-Control group. Whether the mechanisms, in patients with large fractions of non-ventilated lung regions, first and careful attention should be devoted to delivery of low tidal volume.

As mentioned, pulmonary vascular factors could have played a role in the pathogenesis of VILI and NVLI in our model. Redistribution of pulmonary perfusion away from the non-ventilated lung could have also promoted VILI to the ventilated lung. Accordingly, experimental data indicate a synergistic injurious effect of elevated alveolar stretch and overperfusion[35]. However, in the presence of reduced lung stress (NVLI-7.5 group), the role of hyperperfusion in developing VILI was re-dimensioned.

In summary, this preclinical study could advance our current understanding of the development and management of lung injury in mechanically ventilated patients in multiple ways: first, it challenged the concept that non-ventilated areas are automatically protected from injury; second, it highlighted the multifactorial nature of NVLI (hypoperfusion, collapse, high inspiratory pressure in residual ventilated lung, inflammation with endothelial activation), and the relevant interactions between most of these mechanisms measured (e.g., lung stress and inflammation); third, it showed the limits of simply limiting inspiratory lung stress and strain to prevent injury. In terms of treatment, the relative contribution of each underlying factor is yet to be demonstrated, and, by now, conservative approach with equal emphasis on each component is warranted (e.g., reducing tidal volume vs. limiting hypoperfusion).

The clinical implications of our study may be multiple and relevant: (1) Patients undergoing prolonged exclusion of large lung regions from ventilation could be at risk of developing lung injury of the non-ventilated areas, not merely through subsequent reperfusion; (2) The role of regional exclusion from ventilation and hypoperfusion in generating lung injury is neglected and rarely, if ever, measured in clinical practice and could receive more attention and dedicated monitoring (e.g., by dynamic CT scan, EIT); (3) We provided additional evidence on the protective role of low tidal volume ventilation to the lungs, with and without PEEP. However, lowering tidal volumes is still poorly implemented in clinical practice.

While this study provides the "proof-of-concept" for existence, pathophysiology, and main mechanisms of NVLI, the exclusion of one lung from ventilation cannot reproduce the regional heterogeneity of lung

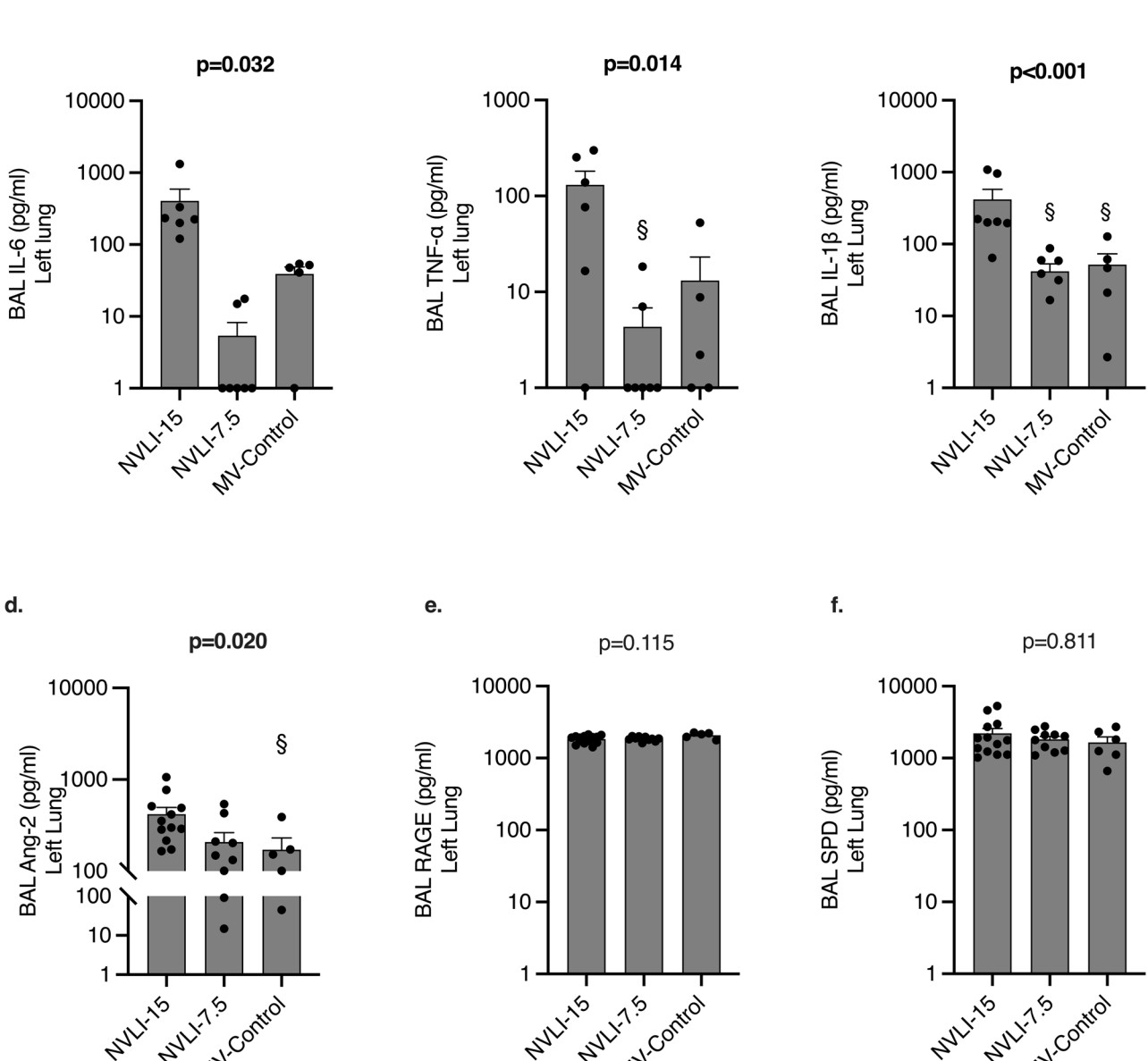

**Fig. 6 | Cytokines concentration from left lung bronchoalveolar lavage (BAL).**
Higher alveolar concentration of the acute phase inflammatory cytokines IL-6 (**a**), IL-1β (**b**) and TNF-α (**c**) were measured in the left lungs of NVLI-15 vs. NVLI-7.5 and MV-Control groups. BAL concentrations of endothelial inflammatory marker Ang2 (**d**) were higher in the left lung on the NVLI-15 group vs. the other 2 groups, while concentrations of epithelial inflammation markers sRAGE (**e**) and SP-D (**f**) didn't differ between groups. Data are expressed as scatter plot with bars and error bars (mean ± SEM). Comparisons were performed by Kruskal-Wallis test for non-normally distributed values (*P*-value reported in the graph) followed by Dunn's multiple comparisons test (§*p* < 0.05 vs. NVLI-15 group). Sample size: NVLI-15 group *n* = 10; NVLI-7.5 *n* = 11; MV-Control *n* = 6.

diseases and the presence of interfaces between ventilated and non-ventilated areas. Moreover, human pathophysiology of NVLI could be characterized by variable amount of non-ventilated regions, presence of "gray" regions (hypoventilated and hypoperfused) and heterogenous distribution of perfusion, which could dampen or multiplicate the observed mechanisms to an unpredictable extent. All these factors, in addition to the relatively short duration of our experiment (hours) compared to the length of mechanical ventilation in critically ill patients (days to weeks), limit the transferability of the study results to the clinical context.

The strengths of our experimental study are: the use of an in vivo model where non-ventilated and ventilated regions coexist but are clearly compartmentalized; the characterization of local injury and inflammation in ventilated and non-ventilated regions; the quantification of local physiologic mechanisms of injury including regional perfusion, stress, and inflammation.

There are also methodological limitations: noticeably, hypoxic pulmonary vasoconstriction is very effective in pigs[36], likely resulting in extreme hypoperfusion (<20% of pulmonary perfusion) of the non-ventilated lung in the two experimental NVLI groups, as compared to patients; we did not measure tissue hypoxia in the non-ventilated lung and we relied on a visual qualitative description of collapse; we did not provide direct evidence of the spread of inflammation between ventilated and non-ventilated areas and vice-versa; finally we didn't analyze plasmatic inflammation.

In conclusion, this experimental study describes that NVLI can develop within 24 h in the non-ventilated lung regions and that lower tidal volume

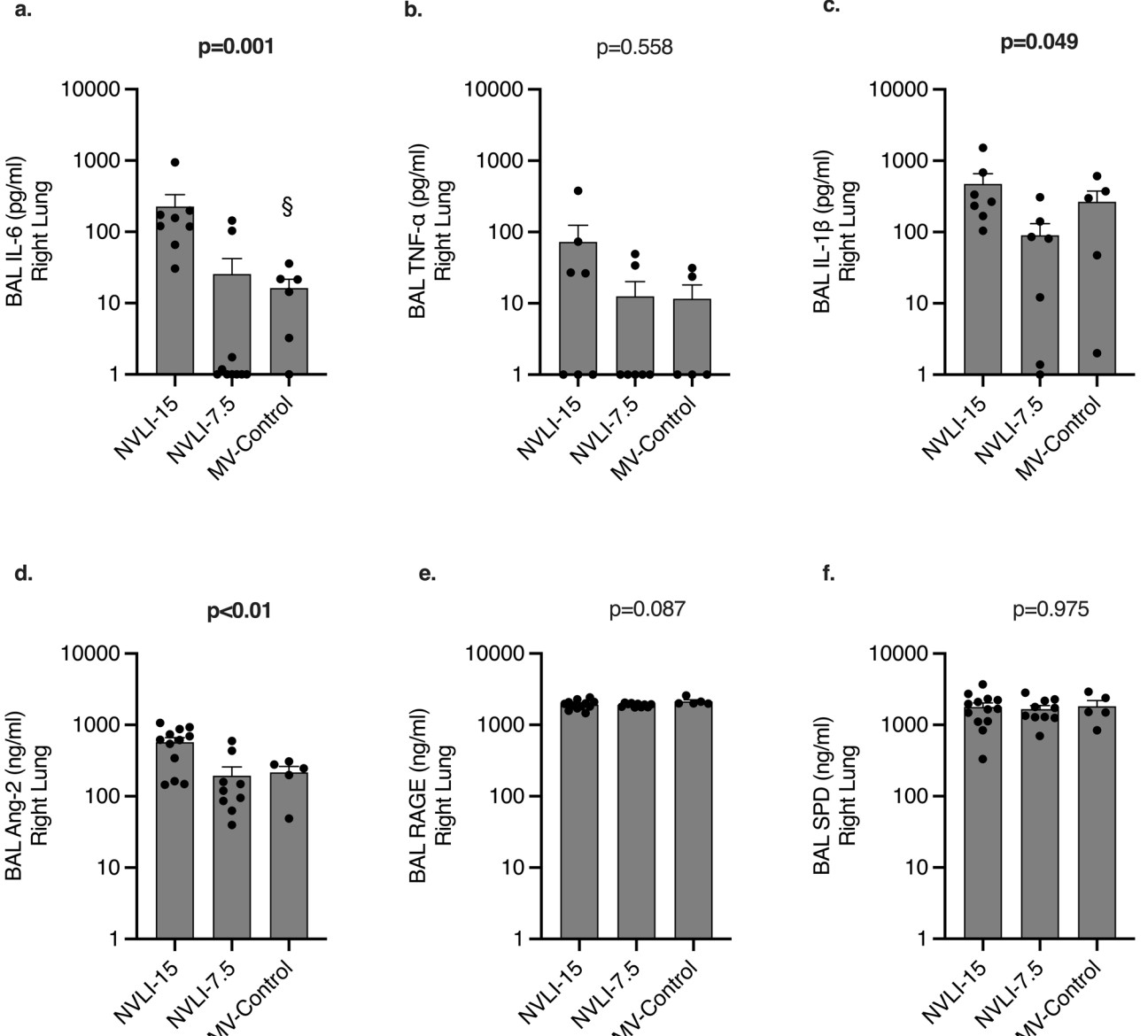

**Fig. 7 | Cytokines concentration from right lung bronchoalveolar lavage (BAL).** Higher alveolar concentration of the acute phase inflammatory cytokines IL-6 (**a**), IL-1β (**b**) and TNF-α (**c**) were measured in the right lung of NVLI-15 vs. NVLI-7.5 and MV-Control groups. BAL concentrations of the endothelial inflammatory marker Ang2 (**d**) were higher in the right lung in the NVLI-15 group vs. the other 2 groups, while concentrations of epithelial inflammatory markers sRAGE (**e**) and SP- D (**f**) didn't differ between groups. Data are expressed as scatter plot with bars and error bars (mean ± SEM). Comparisons were performed by Kruskal-Wallis test for non-normally distributed values (*P*-value reported in the graph) followed by Dunn's multiple comparisons test ($^{\S}p < 0.05$ vs. NVLI-15 group). Sample size: NVLI-15 group $n = 10$; NVLI-7.5 $n = 11$; MV-Control $n = 6$.

provides only partial protection. NVLI results from local factors, such as collapse and hypoperfusion, but is also associated with inspiratory stress to the ventilated lung and inflammatory activation.

## Data availability

All data generated or analyzed during this study are included in this article as Source data (Supplementary Data 2). Source data for Figs. 1, 3, 5–7, Supplementary Figs. 1–2 and Supplementary Data 1 can be found in Supplementary Data 2. All other data are available from the corresponding author on reasonable request.

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

## Acknowledgements

This study was supported, in part, by Current Research from the Italian Ministry of Health, Rome, Italy; by EuroELSO Research grant 2021; by the "Hub Life Science-Diagnostica Avanzata (HLS-DA), PNC-E3-2022-23683266-CUP: C43C22001630001/MI-0117" Project from the Italian Ministry of Health (Piano Nazionale Complementare Ecosistema Innovativo della Salute), Rome, Italy; by the "Dipartimenti di Eccellenza Program 2023–2027" to the Dept. of Pathophysiology and Transplantation, University of Milan, from The Italian Ministry of Education and Research (MUR), Rome, Italy.

## Author contributions

Substantial contributions to the conception and design of the work: ES, AD, FD, SG, TM; acquisition and analysis of data: ES, AD, FD, BG, FO, ZB, ML, MB, CL, ZZ, AMS, GL, AS, TM; interpretation of data for the work: ES, VV, SF, LR, SG, TM. Drafting the work or revising it critically for important intellectual content: all authors. Final approval of the version submitted for publication: all authors.

## Competing interests

The authors declare the following competing interest: TM received personal fees for speaking in sponsored symposia from Drager, Fisher and Paykel, Mindray and Telesair, outside of the submitted work. The other authors declare no competing interests.

## Additional information

[1]Department of Anesthesia, Critical Care and Emergency, Fondazione IRCCS Ca' Granda Ospedale Maggiore Policlinico, Milan, Italy. [2]Department of Pathophysiology and Transplantation, University of Milan, Milan, Italy. [3]Division of Thoracic Surgery and Lung Transplantation, Fondazione IRCCS Ca' Granda Ospedale Maggiore Policlinico, Milan, Italy. [4]Center for Preclinical Research, Fondazione IRCCS Ca' Granda, Ospedale Maggiore Policlinico, Milan, Italy. [5]Furtwangen University, Institute of Technical Medicine, Villingen-Schwenningen, Germany. [6]Division of Pathology, Fondazione IRCCS Ca' Granda Ospedale Maggiore Policlinico, Milan, Italy. [7]Department of Biomedical Surgical and Dental Sciences, University of Milan, Milan, Italy. ✉e-mail: tommaso.mauri@unimi.it

