## [Peer Review File · Communications Medicine]

Reviewers' comments:

Reviewer #1 (Remarks to the Author):

Thank you for the opportunity to review this exciting manuscript. Spinelli et al present the pathophysiological profile of non-ventilated lung injury in healthy pigs undergoing mechanical ventilation. The study was well-designed and hard to perform. I have some comments and questions for the authors regarding the manuscript:

1. The non-ventilated lung model with single lung ventilation replicated by dual-lumen tube was adopted in this experiment, but it is different from the ventilated and non-ventilated lung in one lung in clinical practice. The advantage of this model is that it is easy to replicate, but the problems are also obvious and need to be elaborated in the discussion.
2. Lung injury was significantly caused by large tidal volume and high driving pressure (>24) in the right lung in the NVLI-15 group. Both tidal volume and driving pressure (15 cmH₂O) were in in NVLI-7.5 group at the risk range, but PEEP was 1. and the intrathoracic pressure was Higher in NVLI-15 group, that may cause more lung collapse in left lung. Why not consider abandoning the NVLI-15 group and adding a titrated PEEP group plus the NVLI-7.5 group that protects the ventilated lung?
3. Fig 6 showed that higher alveolar concentrations of the acute phase inflammatory cytokines IL-6 (panel A), IL-1 β (panel B), and TNF- α (panel C) were measured in the left lungs of NVLI-15 vs. NVLI-7.5 and MV-Control groups. However, it appears that the NVLI-7.5 group has the lowest IL-6 levels, please explain appropriately in the discussion.

Reviewer #2 (Remarks to the Author):

This study sheds light on a previously overlooked aspect of mechanical ventilation – the potential for non-ventilated lung injury (NVLI). Traditionally, non-ventilated lung regions have been considered safe from ventilation-associated lung injury, but this study challenges that assumption with compelling preliminary data.

Concerns

- 1: The study was conducted on pigs, which may not perfectly represent human physiology or conditions seen in critically ill patients. Animal models have inherent differences from humans, so the translation of these findings to clinical practice needs to be done cautiously.
- 2: The study had a relatively short duration of mechanical ventilation (24 hours). In clinical practice, patients are often ventilated for longer periods, and the effects of NVLI over an extended duration may differ from those observed in this study.
- 3: The study used unilateral mechanical ventilation, which is not commonly used in clinical settings. Most patients receive bilateral ventilation, so the relevance of these findings to the clinical context may be limited.

Reviewer #1 (Remarks to the Author):

C1. The non-ventilated lung model with single lung ventilation replicated by dual-lumen tube was adopted in this experiment, but it is different from the ventilated and non-ventilated lung in one lung in clinical practice. The advantage of this model is that it is easy to replicate, but the problems are also obvious and need to be elaborated in the discussion.

A1. *We thank the reviewer for this remark. We expanded the discussion related to the study limitations as requested (Page 10):*

"While this study provides the "proof-of-concept" for existence, pathophysiology and main mechanisms of NVLI, the exclusion of one lung from ventilation cannot reproduce the regional heterogeneity of lung diseases and the presence of interfaces between ventilated and non-ventilated areas. Moreover, human pathophysiology of NVLI could be characterized by variable amount of non-ventilated regions, presence of "grey" regions (hypoventilated and hypoperfused) and heterogenous distribution of perfusion, which could dampen or multiply the observed mechanisms to an unpredictable extent. All these factors, in addition to the relatively short duration of our experiment (hours) compared to the length of mechanical ventilation in critically-ill patients (days to weeks), limit the transferability of the study results to the clinical context."

C2. Lung injury was significantly caused by large tidal volume and high driving pressure (>24) in the right lung in the NVLI-15 group. Both tidal volume and driving pressure (15 cmH₂O) were in in NVLI-7.5 group at the risk range, but PEEP was 1. and the intrathoracic pressure was Higher in NVLI-15 group, that may cause more lung collapse in left lung. Why not consider abandoning the NVLI-15 group and adding a titrated PEEP group plus the NVLI-7.5 group that protects the ventilated lung?

A2. *Thanks for your comment. Our study was focused on mechanisms of disease rather than treatments, and the reduction of tidal volume was performed to dissect the effects of lung stress and strain to the ventilated side. However, for a different project, we conducted pilot experiments to explore the effects of lung protective ventilation on NVLI. We applied one-lung volume-controlled ventilation with tidal volume 5-6 ml/kg, respiratory rate to obtain normocapnia and personalized PEEP set by EIT (Jonkmann A. et al. AJRCCM 2023). This was a small (n = 4) exploratory non-randomized group (denominated "NVLI-LPV") performed after completion of the current study. Histological injury score of the left lung in the NVLI-LPV group was equal or higher than NVLI-7.5 (Figure below), further suggesting the marginal protection granted to the non-ventilated lung by protective ventilation. This additional group was not added to the results for methodological limitations and the limited information it would add.*

C3. Fig 6 showed that higher alveolar concentrations of the acute phase inflammatory cytokines IL-6 (panel A), IL-1 β (panel B), and TNF- α (panel C) were measured in the left lungs of NVLI-15 vs. NVLI-7.5 and MV-Control groups. However, it appears that the NVLI-7.5 group has the lowest IL-6 levels, please explain appropriately in the discussion.

A3. We appreciate the careful reading of the study results. We could hypothesize that the lower levels of IL-6 in the NVLI-7.5 group might result from the anti-inflammatory effect of hypercapnia. Indeed, hypercapnia has been shown to decrease the production of inflammatory cytokines via nuclear factor- κ B-dependent mechanism in experimental VILI (Contreras M. et al Crit Care Med 2012). We now clarified this possibility in the discussion (Page 9):

“Finally, permissive hypercapnia in the NVLI-7.5 group could have, at least theoretically, contributed to the partial protection from NVLI [25], possibly also explaining the lower levels of inflammatory cytokines in the NVLI-7.5 as compared to MV-Control group.”

Reviewer #2 (Remarks to the Author):

Concerns

C1: The study was conducted on pigs, which may not perfectly represent human physiology or conditions seen in critically ill patients. Animal models have inherent differences from humans, so the translation of these findings to clinical practice needs to be done cautiously.

C2: The study had a relatively short duration of mechanical ventilation (24 hours). In clinical practice, patients are often ventilated for longer periods, and the effects of NVLI over an extended duration may differ from those observed in this study.

C3: The study used unilateral mechanical ventilation, which is not commonly used in clinical settings. Most patients receive bilateral ventilation, so the relevance of these findings to the clinical context may be limited.

A1-3: We thank the reviewer for appreciating our work. We agree with the limitations indicated by the Reviewer which we added to the limitations section:

"While this study provides the "proof-of-concept" for existence, pathophysiology and main mechanisms of NVLI, the exclusion of one lung from ventilation cannot reproduce the regional heterogeneity of lung diseases and the presence of interfaces between ventilated and non-ventilated areas. Moreover, human pathophysiology of NVLI could be characterized by variable amount of non-ventilated regions, presence of "grey" regions (hypoventilated and hypoperfused) and heterogenous distribution of perfusion, which could dampen or multiply the observed mechanisms to an unpredictable extent. All these factors, in addition to the relatively short duration of our experiment (hours) compared to the length of mechanical ventilation in critically-ill patients (days to weeks), limit the transferability of the study results to the clinical context."

REVIEWERS' COMMENTS:

Reviewer #1 (Remarks to the Author):

The author has answered my question and revised the manuscript.

Reviewer #2 (Remarks to the Author):

It seems the authors have addressed all my concerns. The manuscript significantly improve its potential for acceptance for publication.